# A qualitative exploration of mental health services provided in community pharmacies

Carmen Crespo-Gonzalez[1], Sarah Dineen-Griffin[2], John Rae[1], Rodney A. Hill[3]*

1 School of Dentistry and Medical Sciences, Charles Sturt University, Bathurst, NSW, Australia, 2 School of Biomedical Sciences and Pharmacy, The University of Newcastle, Callaghan, NSW, Australia, 3 School of Biomedical Sciences, Charles Sturt University, Wagga Wagga, NSW, Australia

* rhill@csu.edu.au

**Data Availability Statement:** All relevant data are within the paper and its Supporting Information files.

**Funding:** RAH is the principal recipient and JR and SD-G are co-recipients of funding from the

## Abstract

The burden of mental health problems continues to grow worldwide. Community pharmacists', as part of the primary care team, optimise care for people living with mental illness. This study aims to examine the factors that support or hinder the delivery of mental health services delivered in Australian community pharmacies and proposes ideas for improvement. A qualitative study was conducted comprising focus groups with community pharmacists and pharmacy staff across metropolitan, regional, and rural areas of New South Wales, Australia. Data were collected in eight focus groups between December 2020 and June 2021. Qualitative data were analysed using thematic analysis. Thirty-three community pharmacists and pharmacy staff participated in an initial round of focus groups. Eleven community pharmacists and pharmacy staff participated in a second round of focus groups. Twenty-four factors that enable or hinder the delivery of mental health services in community pharmacy were identified. Participant's perception of a lack of recognition and integration of community pharmacy within primary care were identified as major barriers, in addition to consumers' stigma and lack of awareness regarding service offering. Suggestions for improvement to mental health care delivery in community pharmacy included standardised practice through the use of protocols, remuneration and public awareness. A framework detailing the factors moderating pharmacists, pharmacy staff and consumers' empowerment in mental health care delivery in community pharmacy is proposed. This study has highlighted that policy and funding support for mental health services is needed that complement and expand integrated models, promote access to services led by or are conducted in collaboration with pharmacists and recognise the professional contribution and competencies of community pharmacists in mental health care. The framework proposed may be a step to strengthening mental health support delivered in community pharmacies.

## Introduction

The burden of mental health problems continues to grow worldwide with a 13% rise in mental health conditions and substance use disorders in the last decade [1]. In Australia, mental illness

Pharmacy Guild of Australia (NSW Branch) as part of the NSW Mental Health Community Pharmacy Program. https://www.guild.org.au/about-us The funders had no role in study design, data collection and analysis, decision to publish, or preparation of the manuscript.

**Competing interests:** The authors have declared that no competing interests exist.

is widespread and has substantial impact at the personal, social and economic levels [2]. Four million Australians had a mental or behavioural condition between 2014 and 2015, a figure which increased to 4.8 million in 2017 [3]. It is predicted that almost half of the Australian population will suffer from a mental health problem at some stage in their lifetime [4]. Due to geographic and social isolation, people living in remote and rural areas in Australia have a higher risk of experiencing poor mental health and wellbeing, evidenced by the significantly higher suicide rates compared to metropolitan areas [5, 6]. Importantly, 4.3 million people were supplied with mental health-related medicine in Australia in 2018–2019 [7]. In 2019–2020, $1.4 billion was spent by the Australian Government on benefits for Medicare-subsidised mental health-specific services and $566 million on subsidised mental health-related prescriptions under the Pharmaceutical Benefits Scheme [8].

Primary health care teams are playing a crucial role in the detection, management, and provision of long-term support to people living with a mental illness [9, 10]. The integration of community pharmacists as part of these teams has been supported and encouraged by government agencies and international institutions [11]. In Australia, community pharmacists are often the most accessible healthcare providers with a network of approximately 5,822 pharmacies [12]. Their accessibility has been evident throughout the COVID-19 pandemic, as community pharmacies have remained open providing essential services and access to medicines and health care. This is particularly important in regional and rural communities where timely access to other healthcare professionals and services may be limited. Specifically in 2004, the WHO recognised the importance of as community pharmacists as part of multidisciplinary teams [13] to optimise care for people living with mental illness, and their families [14]. Community pharmacists' role in mental health includes, but is not limited to, the provision of information about conditions and psychotropic drugs, optimising treatment outcomes and quality use of medicines, medication adherence support, screening and early identification, provision of resources, referral to general practice or other services (e.g., hospital emergency services, drug and alcohol rehabilitation facilities) [15]. There is increasing evidence of the positive impact of pharmacists-led interventions have on adherence [16, 17], depression screening [18–21] and medication reviews for community-based mental healthcare consumers [22]. Nevertheless, the research in this area is limited and further studies demonstrating the effectiveness and cost-effectiveness of pharmacists-led interventions in mental health is needed.

In recent years, key pharmacy bodies worldwide have published reports and frameworks highlighting pharmacists' potential roles in mental healthcare. For example, the International Pharmaceutical Federation published a report showing how pharmacists and pharmaceutical organisations positively influence mental health care [23]. The National Health Service (NHS) of England also developed a framework of core mental health competencies for pharmacists [24]. A report published by the United Kingdom's Royal Pharmaceutical Society [25] and a framework by the Pharmaceutical Society of Australia [26] articulate the role of community pharmacists as partners in multidisciplinary teams to support people with mental health problems. These reports also highlighted the barriers to pharmacists becoming partners in mental health care, such as lack of integration, recognition, time and confidence, which aligns with the available literature focused on other professional pharmacy services [27, 28]. However, there is a lack of research focused on pharmacists' perspectives and experiences regarding mental health service delivery in community pharmacies. Considering the perspectives and experiences of stakeholders involved in mental health service provision is essential to understand and identify factors moderating (i.e., facilitating or hindering) the provision of services within a specific setting [29]. Thus, the present study aims to examine the factors that support delivery of mental health services in Australian community pharmacies in New South Wales (NSW) and proposes ideas for improvement, particularly in regional and rural regions.

## Materials and methods

A qualitative study was conducted. Data were collected in two rounds of focus groups conducted virtually via Zoom. A total of eight focus groups were conducted between December of 2020 and June of 2021 (Fig 1). The reporting standards recommended by the consolidated criteria for reporting qualitative research (COREQ) guided the research [30].

### Round One: Initial data collection

The aim of Round One was to derive information regarding the experiences and perspectives of community pharmacists and pharmacy staff regarding the delivery of mental health services in Australian community pharmacies in NSW. Round One consisted of six focus groups divided between December of 2020 and January of 2021.

**Participants' selection and recruitment.**   Participants were recruited using purposive sampling. The following inclusion criteria were applied:

• Community pharmacists (pharmacy owners and employees) and pharmacy staff;

• working in any type of community pharmacies (e.g., chain, independent, banner);

• located in metropolitan, regional and rural areas of NSW.

Ten participants were recruited per group to account for potential non-attendance as recommended in the published literature [31]. Contact details of potential participants were obtained from the Pharmacy Guild of Australia's NSW Branch project team and also via publicly available lists. Potential participants were initially contacted by telephone and email by

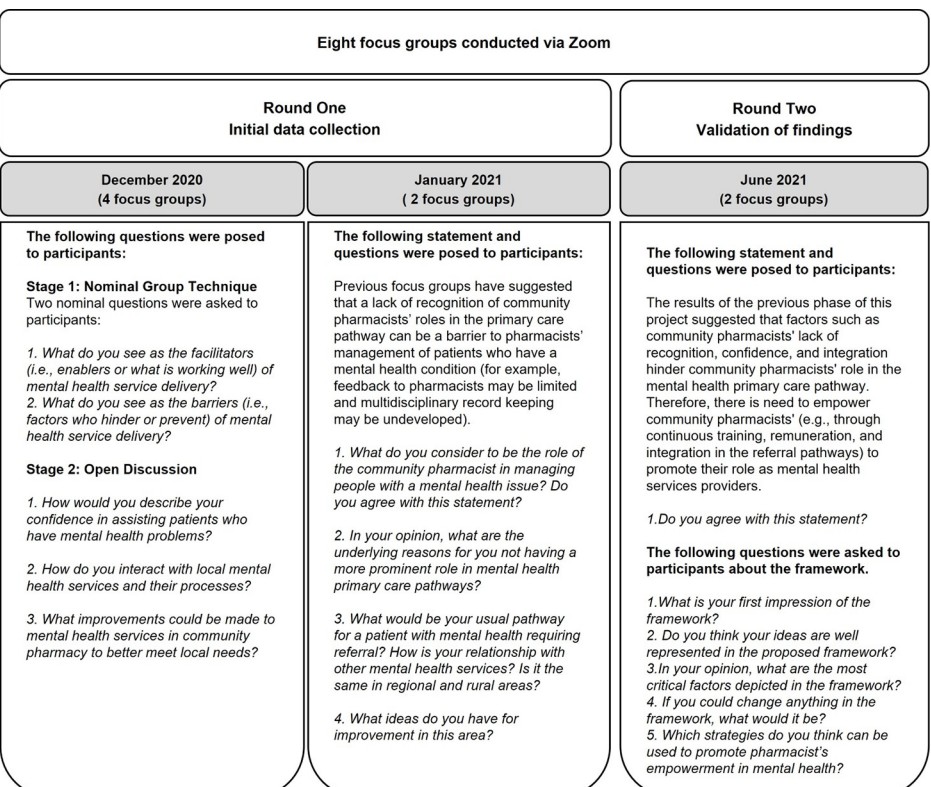

**Fig 1. Study methodology.**

the research team. In addition, the Pharmacy Guild of Australia's NSW Branch project team assisted with contacting eligible participants to participate in the study. Individual pharmacists and pharmacy staff expressing interest in participating were provided with a participant information sheet (PIS) and consent form via email. Demographic data from focus group participants were collected through an online questionnaire developed by the research team. A link to complete the questionnaire was included in the email sent to participants.

**Data collection.** The focus groups were facilitated by two researchers (JR, SDG). An additional researcher was an observer in the group discussion (CCG). A focus group guide (Fig 1) was developed prior to the commencement of the study which was based on previous literature. Data collection was conducted using two different techniques. In December of 2020, data were collected using a combination of the Nominal Group Technique and open discussion questions. In January of 2021, data were collected using open discussion questions.

*December 2020*. Four focus group were conducted. Each focus group meeting was divided into two stages:

*Stage 1*: *Nominal Group Technique*

The Nominal Group Technique (NGT) was applied to explore community pharmacists and pharmacy staff experiences and perspectives [32]. The NGT process is a highly structured face-to-face small group discussion which involves problem solving and generation of ideas which empowers participants by providing an opportunity to have their opinions heard and considered by other participants [33]. The NGT process allows participants to identify issues which require more in-depth exploration and ideas that may not be previously considered by participants [31]. This method was considered appropriate as it limits the influence of the researchers on group participants, enables equal contribution of all group members, allows to generate larger groups of ideas and reach group consensus based on the sum of individual points [34]. It is a time-efficient approach due to its directed process and it produces tangible results [35].

At the commencement of the focus group, one facilitator (JR) described the NGT process to participants who had the opportunity to ask questions. Two nominal questions were asked to participants to explore factors (i.e., barriers and facilitators) of mental health service delivery in community pharmacies (Fig 1).

For each question, the following process was followed [36, 37]:

- **Silent generation of ideas (brainstorming):** The facilitator introduced the question and participants were given time to consider the question and write down their ideas.

- **Round-robin recording of ideas:** Participants were asked to contribute an idea to the discussion. The facilitator (JR) called upon all the participants until all ideas were recorded. Ideas were documented by the other facilitator (SDG) and visible to participants throughout this process.

- **Discussion of ideas:** Participants discussed each idea on the list to clarify its meaning. Similar ideas were combined.

- **Voting:** Once all ideas were obtained, the facilitator (JR) asked participants to vote individually for one-third of the total number of ideas generated. Each participant was asked to vote for the ideas that they believed were most important. The sum of votes was recorded.

*Stage 2*: *Open discussion*

To further explore participants' confidence, relationship with other healthcare professionals and strategies for improvement. The list of questions can be found in Fig 1.

*January 2021*. The remaining two focus groups were conducted as an open discussion to obtain further insights of the information retrieved in the previous focus groups. The

facilitator (JR) posed several questions regarding their role as mental health providers in order to extend the knowledge that arose from focus groups 1–4 (Fig 1).

The focus groups were audio-recorded and transcribed in Zoom. Transcripts were de-identified to ensure privacy and confidentiality and managed in NVivo 12 (QSR International).

**Data analysis.** Thematic analysis of the six focus groups in Round One was conducted by two researchers (JR, CCG) and discussed with the research team (RAH, SDG) to justify participants' ideas obtained during the NGT process and identify additional information resulting from the open discussions [38]. Analysis follows five key stages which included (i) familiarization and initial data coding, (ii) generation of initial codes, (iii) searching for themes, (iv) reviewing themes, and (v) defining themes [39]. All audiotaped files were listened to in order to test the precision of the transcript. A researcher (CCG) read and openly coded the transcripts. The initial codes were reviewed by another researcher (JR) and updated after a discussion between the researchers. Following initial coding, themes were generated to create the whole data image, including the relationships between the themes. The final themes were discussed and approved by the research team. An initial framework was developed following the analysis which depicted the factors that enable or hinder (barriers and facilitators) mental health service delivery in community pharmacy.

## Round Two: Validation of findings

The aim of the second round of focus groups was to validate the findings obtained in Round One and to discuss an initial framework developed following the analysis of data. Two additional focus groups were conducted via Zoom in June of 2021.

**Participants' selection and recruitment.** Participants who participated in Round One of focus groups were re-contacted via email to attend the second round of focus groups. Participants received a new PIS and consent form via email. The goal was to recruit between 4 and up to 10 participants per focus group.

**Data collection.** The focus groups were facilitated by three researchers (JR, SDG, CCG). A focus group guide (Fig 1) and a Framework (Fig 2) were developed based on the results of Round One. Participants were first prompted with a statement and a question for discussion (Fig 1).

Following the discussion, the framework (Fig 2) was presented to the participants as part of the PowerPoint presentation. Several questions related to the framework were asked to participants (Fig 1).

**Data analysis.** A thematic analysis was conducted following the same process as described in Round One.

**Ethics consent and permission.** The Charles Sturt University Human Research Ethics Committee approved this study (HREC number: H20326). All participants provided written consent and received a $50 gift card (per focus group) for their participation.

# Results

## Round One: Initial data collection

**Participants' characteristics.** A total of 40 participants were recruited in Round One. Seven participants dropped out prior to the commencement of the focus groups due to work commitments. The final 33 participants were allocated according to their role in the community pharmacy (i.e., pharmacists or pharmacy staff); the location of the pharmacy (i.e., urban, regional, rural or remote areas) and their certification status (i.e., trained in Mental Health First Aid or not trained) into six groups (Table 1). Most participants were female and the mean age of participants across all groups was 32 years (SD 9.5).

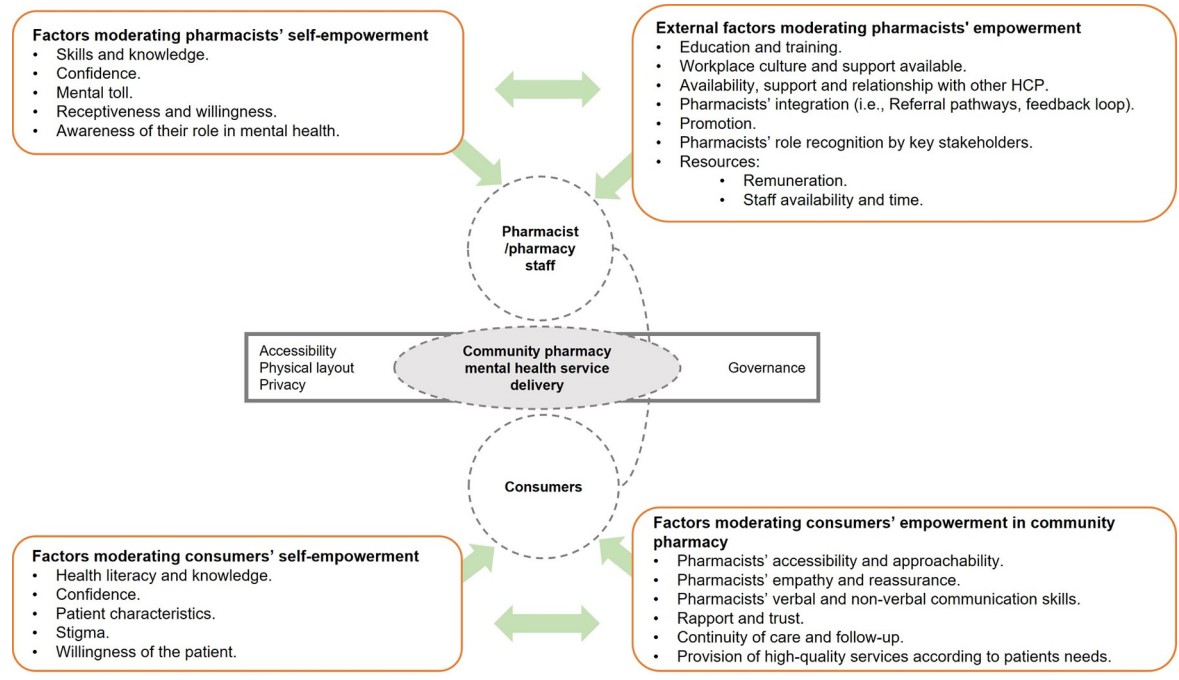

**Fig 2. Framework presented to participants for validation in follow-up focus groups.**

Participant demographics are provided in Table 2. The average duration of the first six focus groups was 89 minutes (SD 7.7).

**Nominal group voting.** From the first four focus groups of Round One, twenty-four factors that enable or hinder mental health service delivery in community pharmacy were identified using the Nominal Group Technique (Tables 3 and 4). The most frequent items identified as facilitators were pharmacists' accessibility, multidisciplinary collaboration, rapport with other healthcare professionals, pharmacists' skills (i.e., knowledge, soft skills and communication skills) and pharmacists' ability to perceive changes of consumers' mental health over time given their frequency of contact. The most common barriers identified were pharmacists' limited time to provide quality mental health services, mental health consumers' stigma and lack

**Table 1. Participants in each focus group.**

| Group | | Number of participants |
|---|---|---|
| A | Community pharmacists located in urban areas, non-trained* (CPUn) | 7 |
| B | Pharmacy staff located in urban areas, non-trained* (PSUn) | 4 |
| C | Community pharmacists located in regional/rural areas, non-trained* (CPRn) | 7 |
| D | Pharmacy staff located in regional/rural areas, non-trained* (PSRn) | 5 |
| E | Community pharmacists located in urban areas, trained (CPUt) | 5 |
| F | Community pharmacists located in regional/rural, trained (CPRt) | 5 |

*Community pharmacists and pharmacy staff who were not Mental Health First Aid[a] (MHFA) certified at the time of the focus group.

[a]Mental Health First Aid: early-intervention courses that increase mental health literacy and teach the practical skills needed to support someone experiencing a mental health problem, experiencing a worsening of an existing mental health problem or in a mental health crisis until appropriate professional help is received or the crisis resolves [40].

**Table 2. Participants' demographics (Rounds One and Two).**

| Characteristics | Round One | | | Round Two | | |
|---|---|---|---|---|---|---|
| | All participants n (%) | Community pharmacists n (%) | Pharmacy staff n (%) | All participants n (%) | Community pharmacists n (%) | Pharmacy staff n (%) |
| Total | 33 (100) | 24 (72.7) | 9 (27.3) | 11 (100) | 9 (72.7) | 2 (18.2) |
| Country of birth | Round One | | | Round Two | | |
| Australia | 27 (81.8) | 21 (63.6) | 6 (18.1) | 7 (63.6) | 6 (54.5) | 1 (9.1) |
| Other (e.g., China, England) | 6 (18.2) | 3 (9.1) | 3 (9.1) | 4 (36.4) | 3 (27.3) | 1 (9.1) |
| Gender | Round One | | | Round Two | | |
| Female | 30 (90.9) | 22 (66.6) | 8 (24.2) | 11 (100) | 8 (72.7) | 3 (27.3) |
| Male | 3 (9.1) | 2 (6.1) | 1 (3.0) | 0 | 0 | 0 |
| Age (years) | Round One | | | Round Two | | |
| 18–24 | 7 (21.2) | 2 (6.1) | 5 (15.2) | 3 (27.3) | 1 (9.1) | 2 (18.2) |
| 25–34 | 19 (57.5) | 17 (55.5) | 2 (6.1) | 6 (54.5) | 6 (54.4) | 0 |
| 35–44 | 4 (12.1) | 3 (9.1) | 1 (3.0) | 1 (9.1) | 1 (9.1) | 0 |
| 45–54 | 1 (3.0) | 0 | 1 (3.0) | 0 | 0 | 0 |
| 55–64 | 1 (3.0) | 1 (3.0) | 0 | 0 | 0 | 0 |
| 65–74 | 1 (3.0) | 1 (3.0) | 0 | 1 (9.1) | 1 (3.0) | 0 |
| Employment | Round One | | | Round Two | | |
| Full-time | 21 (63.6) | 17 (51.5) | 4 (12.1) | 4 (36.4) | 4 (36.4) | 0 |
| Part-time | 6 (18.2) | 1 (3.0) | 5 (15.2) | 3 (27.3) | 1 (9.1) | 2 (18.2) |
| Casual | 4 (12.1) | 3 (9.1) | 1 (3.0) | 3 (27.3) | 3 (27.3) | 0 |
| Not working (i.e., studying or maternity leave) | 2 (6.1) | 1 (3.0) | 1 (3.0) | 1 (9.1) | 1 (9.1) | 0 |
| Highest level of education | Round One | | | Round Two | | |
| Year 12 or equivalent | 6 (18.2) | 0 | 6 (18.2) | 1 (9.1) | 0 | 1 (9.1) |
| Bachelor's degree | 7 (21.2) | 4 (12.1) | 3 (9.1) | 3 (27.3) | 2 (18.2) | 1 (9.1) |
| Graduate diploma/ certificate | 8 (24.2) | 8 (24.2) | 0 | 2 (18.2) | 2 (18.2) | 0 |
| Postgraduate degree | 12 (36.4) | 12 (36.4) | 0 | 5 (45.5) | 5 (45.5) | 0 |
| Pharmacy location | Round One | | | Round Two | | |
| Urban | 16 (48.5) | 13 (39.4) | 3 (9.1) | 7 (63.6) | 6 (54.5) | 1 (9.1) |
| Regional, rural or remote | 17 (55.5) | 11 (33.3) | 6 (18.2) | 4 (36.4) | 3 (27.3) | 1 (9.1) |
| Type of community pharmacy | Round One | | | Round Two | | |
| Independent | 15 (45.5) | 13 (39.4) | 2 (6.1) | 7 (63.6) | 6 (54.5) | 1 (9.1) |
| Banner Group* | 13 (39.4) | 7 (21.2) | 6 (18.2) | 2 (18.2) | 1 (9.1) | 1 (9.1) |
| Discount Chain | 4 (12.1) | 1 (3.0) | 3 (9.1) | 1 (9.1) | 1 (9.1) | 0 |
| Hospital pharmacy | 1 (3.0) | 1 (3.0) | 0 | 1 (9.1) | 1 (9.1) | 0 |
| Number of employees working on an average shift in the community pharmacy | Round One | | | Round Two | | |
| 2 or less | 0 | | | 0 | | |
| 3–5 | 10 (30.3) | | | 5 (45.5) | | |
| 6–8 | 7 (21.2) | | | 2 (18.2) | | |
| 9–11 | 10 (30.3) | | | 1 (9.1) | | |
| 11 or 13 | 3 (9.1) | | | 0 | | |
| 14 or more | 5 (15.2) | | | 3 (27.3) | | |

*Banner group: pharmacies that act as a franchise for marketing, management and purchasing purposes.

**Table 3. Factors identified as facilitators of mental health service delivery in community pharmacy.**

| Factor | Number of groups where this facilitator was identified (n = 4) | Number and (%) of participants voting on this facilitator (n = 33) | Participants located in urban or regional/remote areas. |
|---|---|---|---|
| Accessibility (e.g., first port of call, timeliness of service at no cost, no waiting times) | 3 | 16 (48.5) | All |
| Having good rapport with local general practitioners (GPs), multidisciplinary collaboration, and integration within referral pathways | 3 | 14 (42.4) | All |
| Pharmacists' skills (specialist nature of the pharmacist), knowledge, soft and communication skills (i.e., verbal and non-verbal) | 3 | 13 (39.4) | All |
| Ability to track perceived change of consumers mental health over time (i.e., recognise deterioration or improvements) and frequency of contacts (i.e., monthly at pharmacy), reinforcement following interactions with specialist physicians, and GPs | 3 | 9 (27.3) | All |
| Less formal/ less clinical setting—reducing stigma/ confrontation for people living with a mental health condition | 3 | 8 (24.2) | All |
| Pharmacy physical layout—is it an appropriate space to discuss mental health? | 3 | 2 (6.0) | All |
| Rapport, trust and consistency in community pharmacist-consumer relationship (compared with other primary care providers) | 2 | 8 (24.2) | Regional/Rural |
| Employees and pharmacists—ability to ensure best services are provided, priority and referral (connecting people with appropriate services) | 2 | 8 (24.2) | All |
| Approachability of staff (and mix of staff i.e., different cultures, gender), consumer feels comfortable to have these discussions | 2 | 4 (12.2) | All |
| Opportunity to engage and follow up a person every time community pharmacists dispense a medication for mental health (i.e., monthly, or for opioid treatment program consumers (daily/ weekly) | 2 | 3 (9.1) | Urban |
| Time—more pharmacists on duty to deliver professional services | 2 | 3 (9.1) | All |
| Knowledge and understanding of other/ network mental health services in local area/ referral patterns | 2 | 2 (6.0) | All |
| Other services—such as home medicines delivery, can check on mental health consumer | 2 | 0 | All |
| Staff training available to pharmacists/ staff | 1 | 4 (12.2) | Urban |
| Up-to-date knowledge and education for pharmacists and staff (i.e., clinical knowledge) | 1 | 3 (9.1) | Urban |
| Pharmacists' ability to acknowledge and respect boundaries between themselves and consumers | 1 | 2 (6.0) | Urban |
| Healthy workplace culture (i.e., for pharmacists and staff) | 1 | 1 (3.0) | Urban |
| Promoting role of pharmacist in mental health | 1 | 1 (3.0) | Urban |
| Scope of practice—knowing staff/ pharmacist limitations when managing/ helping a person living with a mental illness | 1 | 1 (3.0) | Urban |
| Implementing a system to document consumer interactions | 1 | 0 | Urban |
| Consumer resources (leaflets, phone numbers, direct referrals) | 1 | 0 | Urban |
| Willingness of pharmacists to use technology and adapt | 1 | 0 | Urban |
| Using dispense techs and pharmacy assistants during process (i.e., flagging during dispensing such as change in medicine) | 1 | 0 | Urban |
| Referral made by pharmacist to GP through use of technology (i.e., MedAdvisor-extra service) | 1 | 0 | Regional/Rural |
| Staff safety | 1 | 0 | Urban |

**Table 4. Factors identified as barriers for the delivery of mental health services in community pharmacy.**

| Factor | Number of groups where this barrier was identified (n = 4) | Number and % of participants voting on this barrier (n = 33) | Participants located in urban or regional/remote areas. |
|---|---|---|---|
| Time for pharmacists and pharmacy staff to communicate effectively | 4 | 15 (45.5) | All |
| Stigma of consumers around having mental health problem | 4 | 14 (42.4) | All |
| Pharmacists support from employers | 3 | 13 (39.4) | All |
| Lack of training/ experience to address stigma/ communication skills to have the confidence to have harder conversations | 3 | 5 (15.2) | All |
| Limited mental health services available | 3 | 5 (15.2) | All |
| Privacy issues | 3 | 3 (9.1) | All |
| Remuneration—inconsistent services provided across pharmacies given the lack of remuneration associated with the service/ lack of quality indicators | 2 | 9 (27.3) | All |
| Confidence of the pharmacist to deliver the mental health service (high risk/complex cases) | 2 | 7 (21.2) | Regional/Rural |
| Standardised process to practice and processes to clinically document interactions with consumers, in same system other health providers operate | 2 | 6 (18.2) | All |
| Pharmacists' and pharmacy staff receptiveness and beliefs (pharmacists may not want to advance in this area of practice) | 2 | 4 (12.2) | All |
| Gaps in continuity of care between multiple health providers, especially in rural or remote areas | 2 | 3 (9.1) | Regional/Rural |
| Promotion of role of pharmacist (i.e., consumers and other health care providers unaware of what services can be offered in community pharmacy) | 2 | 2 (6.0) | All |
| Language and cultural barriers (special populations) | 2 | 1 (3.0) | All |
| Physical layout | 2 | 0 | All |
| Scope of practice (pharmacists first port of call/ initial contact) and knowing where we fit in to broader care process | 2 | 0 | All |
| Pharmacists forgotten in primary care pathways/ no direct access to referral pathways | 1 | 6 (18.2) | Urban |
| Quality of service consistency because of pharmacy support available/ acknowledged tension from employers regarding priorities | 1 | 5 (15.2) | Urban |
| Low consumer health literacy (i.e., their understanding of their condition, medicines and other areas of management) | 1 | 4 (12.2) | Urban |
| Lack of free support/ free training for pharmacy staff | 1 | 3 (9.1) | Regional/Rural |
| Mental toll it takes on pharmacy staff following interactions with people living with a mental health condition | 1 | 3 (9.1) | Regional/Rural |
| Consumers tired of repeating their information to multiple health providers, availability of information across the network | 1 | 2 (6.0) | Regional/Rural |
| Financial barriers, consumer out-of-pocket costs | 1 | 2 (6.0) | Urban |
| Consumers' willingness to change behaviour and perceived severity of their condition | 1 | 1 (3.0) | Urban |
| Staff safety | 1 | 1 (3.0) | Urban |
| Consumers' trust | 1 | 0 | Urban |
| Consumers' time | 1 | 0 | Regional/Rural |
| Setting/ location of pharmacy | 1 | 0 | Regional/Rural |
| Physical accessibility of the pharmacy (e.g., consumers with disability) | 1 | 0 | Regional/Rural |

of support from employers for the provision of mental health services given the focus on the delivery of remunerated services.

Rapport with consumers in regional and rural areas was identified as one of the most critical enablers of mental health service delivery in community pharmacy. Education and training were considered essential by community pharmacists in urban areas. Pharmacists' lack of confidence to provide these services was frequently identified as a barrier to mental health service delivery by rural pharmacists and pharmacy staff. Community pharmacists in urban areas highlighted the lack of recognition and integration of community pharmacists in primary care referral pathways.

**Thematic analysis.** The referral pathways used by community pharmacists and pharmacy staff in urban and regional/rural areas have been identified through the thematic analysis and the data obtained in the six-focus groups of Round One. Strategies for improvement for mental health training of community pharmacists have been gathered, and a framework to guide pharmacists' and consumers' empowerment for the delivery of mental health services in community pharmacy has been proposed.

*Referral pathways used by community pharmacists in urban, regional and rural areas.* Most participants described general practitioners (GPs) as the primary point of referral for consumers with a mental health condition.

> CPRt1: *"The first point of referral is definitely the GP."*

> CPUt1: *"The best contact that I've had in my community pharmacy career is just with GPs."*

Participants also declared that when more severe cases were detected, they referred consumers to a mental health specialist.

> CPUt6: *"If someone's had a crisis and is very suicidal, basically all you can do is call the ambulance to get them to the hospital in a crisis."*

> CPRt3: *"In Orange, there's a mental health facility. . .they will also have psychiatrists down there as well, so the mental health nurses will try and fit them in down there if they need to be seen"*.

In cases in which consumers were identified to be dealing with milder mental illness, participants reported that they provided information to their consumers about the availability of free resources and other services, such as helplines.

> CPMn1: *"If it's mild mental illness. . .you can refer the consumer to online services."*

> CPRt2: *"[Referral to] online services and cognitive behaviour therapy services. . .we use a lot of because. . . psychiatrists and psychologists. . .there's a huge waitlist for most of them."*

> CPUn1: *"Making aware to them [consumers] the options that are available over the phone or on the internet, things like that that are more anonymous."*

*Strategies for improvement.* Most participants in both metropolitan and regional/rural groups commented on the necessity of including additional training more focused on pharmacy-specific real-life scenarios, multidisciplinary learnings and pharmacists' soft skills (e.g., emotional intelligence, empathy, verbal a non-verbal communication).

> PSRn3: *"With the company that I'm with, they do actually provide module training which has been helpful, but I don't really think it touches on as much as you need to feel confident."*

*CPUt6: "Particularly with mental health first aid, there were no pharmacy specific scenarios, and pharmacy is a very unique environment to be applying these."*

*CPUt5: "I think you first need some examples of real-world scenarios. . .and the opportunity to actually attempt to apply it [in practice]. . .have someone to help you talk through what happened, what went wrong, what worked, what didn't."*

Participants in both metropolitan and regional/rural groups indicated their willingness to receive mental health training (i.e., those who had not previously undertaken the Mental Health First Aid course). Those who had previously undertaken Mental Health First Aid training suggested a refresher course to be completed every two to three years. Some participants highlighted the need to standardise pharmacist mental health service delivery through guides and standard documentation for consistency in practice.

*CPRn2: "The key is consistency, just making something uniform. . . like we have our S3 [Schedule 3] protocols. . .should we have a mental health protocol to address the situation and highlight red flags that need immediate referral. . .that could be a way forward to help with consistency."*

*CPUn4: "If you're able to standardise practice and the service that you're providing. . .I suppose processes that you're able to quantify. . .you're able to understand how much time is required and therefore how much cost is associated."*

Community pharmacists located in urban areas identified the need to improve government recognition of the role of community pharmacists in mental health in order to enhance the integration of community pharmacy in primary care.

*CPUn1: "Better understanding of other primary care services already provided in the community because. . .you understand where the need is, and you'll be able to see what you can fit in your practice."*

*CPUt5: "We need increased recognition if a pharmacist is going to be an active part in mental health screening or planning. We need differentiation and it shouldn't be something that's readily available."*

Community pharmacists and pharmacy staff in rural and regional areas expressed the need to promote pharmacists' roles (e.g., signage, advertising) and the mental health services pharmacies offer.

*CPUn5: "On radio. . .you know, let local people know they have someone to talk about their mental health and they can go see a pharmacist. . .this is what a pharmacist can do."*

*CPRn4: "Try and raise consumer awareness of how accessible we [pharmacists] are. The [Pharmacy] Guild did it a couple of years ago with the 'Ask your pharmacist' campaign, but if that could actually be a bit more with a mental health perspective? Maybe the public don't think of us as somebody who can help if they're having mental health issues".*

Participants in all groups highlighted the necessity of funding to facilitate pharmacists' provision of mental health services to their community. The possibility of including a remunerated medication review focused on mental health was also suggested.

*CPUn6: "I think one of the big things is funding. We need some way to get paid for our service as a pharmacist. . .to be able to claim an appointment fee. . .to spend even 10 minutes with a consumer in the room."*

*Initial proposed framework for pharmacist, pharmacy staff and consumer empowerment in mental health.* Thematic analysis of the data obtained in the six focus groups of Round-One to further explore factors moderating mental health service delivery was conducted. A full list of the factors and supporting quotes can be found in S1 Appendix. The information retrieved highlighted pharmacists and pharmacy staff perception of a lack of support, recognition and integration within primary care referral pathways as major barriers to delivering mental health services in community pharmacy. Consumer stigma regarding mental health and their lack of awareness around service offering were also identified as a barrier from pharmacy and pharmacy staff point of view. As a result, a framework has been developed detailing factors to consider which can enable or hinder pharmacists' and consumer empowerment in mental health care delivery in community pharmacy (Fig 2).

## Round Two: Validation of findings

**Participants' characteristics.** Sixteen participants were recruited in Round Two. Five of them dropped out before the beginning of the focus groups due to work commitments. Eleven participants (i.e., community pharmacists and staff located in urban and rural/regional areas) were allocated to the two follow up focus groups to validate the findings from Round One. Participant demographics are provided in Table 2. The average duration of the two focus groups was 86 minutes (SD 17.6).

**Thematic analysis.** Most participants agreed that there is a need to empower community pharmacists to promote their role as mental health providers for example, through continuous training:

*CPU1: "I completely agree with this statement. . .how it's phrased and I like the idea of empowering. . .not just educating pharmacists. . .it's empowering them to take a stance and be more involved in the mental health aspect of their consumers.."*

*CPR2: "I agree with the last bit, especially that we need to empower ourselves as mental health services providers. . .it is something that comes up very often in pharmacy and it's something that requires training because it's not easy to do if you don't know where to start or how to start that conversation with people."*

One of the participants pointed out that many aspects need to be considered and addressed to empower community pharmacists and pharmacy staff.

*CPU5: "The main thing that stands out for me is in terms of empowering is that has to be multifaceted. . .it can't be just through training and it can't be just renumeration, it's going to have to encompass all of those things."*

In some cases, participants reported that community pharmacists and pharmacy staff already have the skills to respond to consumers with mental health issues. Still, the lack of integration within the health system hindered achieving this goal.

*CPR5: "I understand what you're trying to say by empower, but I feel, in some cases it's not about the pharmacists not having the power to do it, but it's more like enabling them to be*

*integrated as well" "The pharmacist role needs defining in this case as well because you're talking about it being limited and then to promote it, but then what is the role of the pharmacist in light of the whole system? You know what I mean? It's not just what one pharmacy thinks it should be, or what one area but the whole, we need to have consensus, so we don't have to keep on you know validating the role of the pharmacist."*

Participants' suggestions to improve the proposed framework were to establish some sort of priority to identify the first steps to consider enhancing the delivery of mental services in community pharmacies.

*CPR5: "Yeah, I would yeah really love to see like a priority."*

*CPR4: "We did identify a range of factors and I presume that a lot of those may carry more weight than others, so in terms of priority for what you might begin to address in service implementation, it's not clear to me."*

*CPU2: "I think I would like to see remuneration stand out more in the model."*

Participants were asked about strategies to promote pharmacists' and staff empowerment in mental health, they suggested promoting pharmacists' role, integrating and linking community pharmacists with other healthcare professionals.

*CPU1: "Definitely more publicity about what a pharmacist can do, and maybe I personally think like the successful story on how you know my experience have helped consumers and from the consumer perspective it's a very powerful message to other people."*

*CPR2: "Linking pharmacists with the other healthcare providers in the region that are providing mental health services so that they can network properly and get to know, for example, the mental health team unit that might be working in their local area so that you know you've got a face and a name to somebody who might be the clinical case managers to your consumers."*

As in previous focus groups, some participants reinforced the possibility of including medication review services or MedsCheck (i.e., services consisting in a review of a consumer's medicines to improve their understanding of their medicines and ultimately, their health outcomes) for consumers dealing with mental health services.

*CPU2: "Presuming like we're talking about implementing mental health services into Community pharmacy but recognising that we already have an existing suite of various services that we provide around medication management review, I'm talking about things like MedsCheck, so looking at how we can integrate and align any type of service provision with any existing services as well, so we don't have an unrelated set of services."*

*CPR3: "It could be something so simple, such as we've already got like, for example, a diabetes MedsCheck, we can have a mental health needs check something like that."*

When participants were introduced to the factors identified in the first round of focus groups, they highlighted governance as a first step to have organisational support to guide and define the provision of mental health services in the community pharmacy. The importance of consistency in providing quality services within community pharmacies to ensure consumers receive the same type of services and other healthcare professionals recognise the value of community pharmacists' role was also pointed out by participants in both groups. Having support

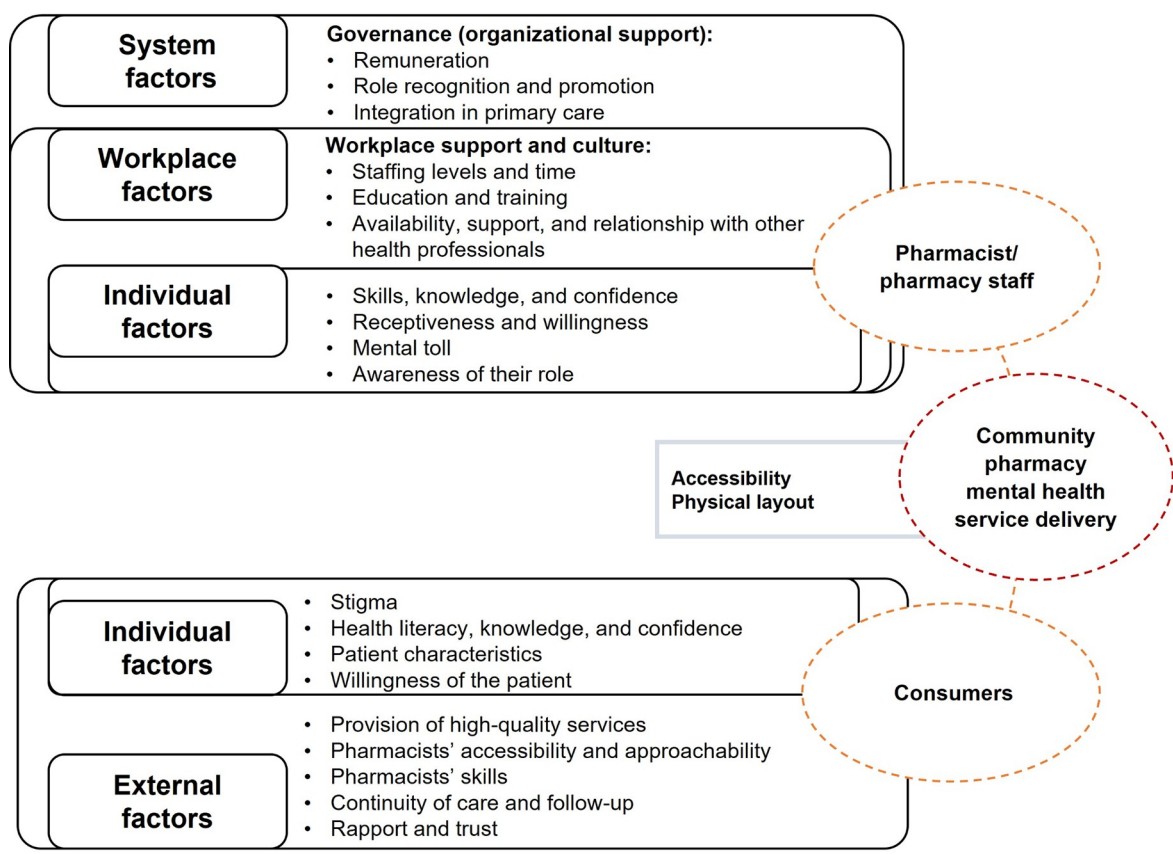

**Fig 3. Framework for pharmacists, pharmacy staff and consumers' empowerment in mental health.**

within the pharmacy workforce, at the workplace, and adequate staffing levels was highlighted by participants and, in some instances, linked to the provision of quality services. Mental health services remuneration and having time allocated to provide the services were the most significant factors highlighted by participants in both groups. Pharmacists' continuous education and training were also highlighted as an essential factor to empowerment in mental health. Supporting quotations can be found in S2 Appendix.

*Final proposed framework for pharmacists*, *pharmacy staff and consumers' empowerment in mental health*. As a result of the information retrieved from the follow-up focus groups, a final framework to enhance pharmacists, pharmacy staff and consumers' empowerment in mental health has been proposed (Fig 3). A full description of the factors included in the framework can be found in Table 5.

## Discussion

Universally, greater stressors impact us in every life aspect in forms of occupational, social, cultural and health imposts. Primary health care at the interface of communities has become much more important and supporting mental health care is perhaps one of the greatest challenges our modern society has faced. The role of community pharmacists and pharmacies has become substantially more important in providing primary mental health support to communities. In this critical context, the present study explores factors (that enable or hinder) the delivery of mental health services in community pharmacies. Referral pathways and strategies for improvement have also been identified.

**Table 5. Description of factors included in the framework.**

| Factors moderating community pharmacy mental health service delivery | | |
|---|---|---|
| Factor | Description | Example |
| Accessibility | Ensure community pharmacies have resources in place to ensure easy access to the community regardless of their situation | Consumers using a wheelchair may require a ramp to access the community pharmacy. |
| Physical layout (privacy) | Having a designated area to provide mental health services to ensure consumers' feel safe to open up about their conditions. This factor has more impact in smaller areas (i.e., rural/regional) where there is less privacy | Pharmacies with counselling rooms may help to provide more comprehensive services to consumers, and they may feel safe. |
| **Factors moderating community pharmacist and pharmacy staff empowerment in mental health** | | |
| System Factors | Description | Example |
| Governance (Organisational support) | Having the set of processes, regulations, policies, and resources to define, regulate and standardise mental health services delivery in community pharmacies to enhance safety, reliability, and quality of care. | Framework defining the role of community pharmacists in mental health care to ensure there is consistency in practice and approach to care by community pharmacies. Guidelines and protocol to support the provision of quality mental health services |
| Remuneration | Community pharmacists/ staff being paid for the time allocated to provide quality mental health services | Lack of funding may prevent community pharmacists/ pharmacy staff from allocating a specific time to provide quality mental health services |
| Pharmacists' role recognition by key stakeholders | Community pharmacists'/ staff role in mental health being defined and recognised by governments, consumers, and healthcare providers | Government and consumers may not understand or recognize how pharmacists/ pharmacy staff can help consumers with mental health |
| Promotion | Availability of resources promoting the role of community pharmacists/ staff in mental health | Mental health services provided in community pharmacies may be advertised through campaigns to increase people awareness of their services |
| Pharmacists' integration (i.e., referral pathways, feedback loop) | Having access to consumers' mental health information in documentation or through personal contact with other primary care providers. Availability of defined referral pathways | Feedback to pharmacists may be limited and multidisciplinary record keeping may be under-developed |
| Workplace factors | Description | Example |
| Workplace support and culture | Community pharmacists' and staff 'goals and priorities aligned. Team collaboration. Encouragement and support from employers | Pharmacists/ pharmacy staff discuss and set priorities and strategies to approach consumers dealing with mental health issues. |
| Staffing levels and time | Community pharmacists' and staff having the chance to allocate time to the provision of mental health services. The staffing levels have a direct impact on time | Small pharmacies may only have one to two employees which may hinder the provision of the service |
| Education and training | Community pharmacists' and staff having the resources and support to upskill in mental health | Pharmacists/ pharmacy staff may have information and access to specific training focused on mental health in their community pharmacy |
| Availability, support, and relationship with other health professionals | Community pharmacists/ staff having access to other primary care providers in their area. The access to primary care providers in remote areas is more limited than in urban areas | In rural areas, access to other mental health services may be limited. However, it may be easier to establish a close relationship with other health professionals |
| Individual factors | Description | Example |
| Skills, knowledge, and confidence | Community pharmacists'/ staff having the required tools to deliver quality mental health services to consumers | Pharmacists/ pharmacy staff communication, active listening, reflection, empathy. Capacity to understand and meet people's health literacy needs and recognize what consumers are taking and why |
| Receptiveness and willingness | Community pharmacists/ staff recognizing the importance of mental health and being willing to help consumers | Pharmacists and staff may not have initiative and motivation due to the lack of time or support |
| Mental toll | Community pharmacists'/ staff having access to resources to support their mental health | Pharmacists/ pharmacy staff may have encountered a problematic consumer dealing with a mental illness, which may provoke the pharmacist to feel unsafe |
| Awareness of their role in mental health | Community pharmacists'/ staff understating their role in mental health | Integration with other mental healthcare providers may be limited, preventing community pharmacists from understanding their mental health role |
| **Factors moderating consumers' empowerment in mental health** | | |
| External Factors | Description | Example |

*(Continued)*

**Table 5.** (Continued)

| | | |
|---|---|---|
| Provision of high-quality services according to consumer's needs | Community pharmacists/ staff assessing and helping consumers according to their situations | Community pharmacists/ pharmacy staff talking with their consumers may identify the consumer does not have the financial resources to seek appropriate care |
| Pharmacists' accessibility and approachability | Easy access to community pharmacists without appointment in a less clinical setting | Consumers in certain areas may not have access to other healthcare providers |
| Pharmacists' skills | Community pharmacists and staff' knowledge, empathy, reassurance, verbal, and non-verbal communication | Community pharmacists/ pharmacy staff may have the opportunity to make the consumers feel understood and normalise their condition |
| Continuity of care and follow-up | Community pharmacists and staff' frequency of interactions with consumers | Pharmacists/ pharmacy staff may see a consumer once or twice a week and can identify if a consumer is coping with their mental health |
| Rapport and trust | Community pharmacists/ staff establishing close relationships with consumers | Pharmacists/ pharmacy staff establishing a close relationship with a consumer may help to increase their confidence and willingness to talk about their condition |
| **Individual factors** | **Description** | **Example** |
| Stigma | Consumers' own ideas regarding their mental health conditions | A consumer arrives at the pharmacy and feels judged by other consumers |
| Health literacy, knowledge, and confidence | Consumers' having the resources to fully comprehend their conditions | A consumer may have poor health literacy level to understand their condition |
| Consumer characteristics | Consumers' demographic characteristics such an age, race and background | A consumer arrives at the pharmacy, his first language is not English, and the pharmacy staff cannot communicate effectively with him |
| Willingness of the consumer | Consumers' fully aware of the situation and wanting to receive help | A consumer who is aware of their mental health condition but is not willing to receive help |

The results of this study have shown that there are multiple barriers (e.g., lack of governance, remuneration) and facilitators (e.g., pharmacists' skills, accessibility) that are crucial to consider for promoting the role of community pharmacists as mental health providers in primary care. Pharmacists' accessibility was identified as the most influential facilitator of mental health service delivery. Community pharmacists are at times the first point of contact for consumers who seek help for mental health conditions, especially in the current circumstances with limited access to other health professionals [41]. This is important for pharmacies operating after hours or in regional/rural communities where access to care may be limited.

Pharmacists' specific skills and drug knowledge were highlighted key facilitators in providing quality mental health care. The positive clinical impact of pharmacists' role in mental health has been previously demonstrated in the literature [42–44]. Consequently, another study found that pharmacist-led mental health adherence interventions for consumers with type 2 diabetes significantly improved psychotropic medication adherence in adult consumers [45].

Consumer willingness to receive a mental health service was identified as a facilitator of mental health service delivery. In a study evaluating consumers' experience in a community pharmacy mental health program, participants recognised community pharmacists' positive influence on their mental health and well-being [46]. However, as reported in this study, the stigma of people living with a mental health issue discourages individuals from getting proper mental health treatment. These results are consistent with literature indicating that the mental health stigma of consumers and mental health services providers has been shown to be a barrier to the effective management of mental health [21, 47, 48].

Training and education in mental health were also reported as facilitators for service provision. Specifically, the importance of training to address stigmatising beliefs and stereotypes has been highlighted [49]. Indeed, mental health training has been shown to impact pharmacists'

confidence positively [50, 51]. Particularly in a study participants indicated comfortability discussing mental illness with community pharmacists trained in Mental Health First Aid (MHFA), revealing an opportunity for pharmacists to expand access to mental health services by being trained in MHFA and counselling about mental illness [52]. Thus, upskilling the pharmacy workforce in mental health should be prioritised.

Participants located in rural and regional areas highlighted rapport with consumers as a facilitator for the provision of mental health services in their communities. Building close relationships with consumers may positively affect their willingness to openly discuss their condition [53]. Collaboration with other primary care providers was also identified as important for the provision of mental health care. The positive influence that pharmacists' collaboration with other primary care providers has on consumers' outcomes has been demonstrated [54, 55]. Adopting a collaborative approach to mental health care has been shown to promote the efficiency and effectiveness of services by sharing healthcare providers' knowledge and skills [56].

Governance (across the pharmacy profession) was highlighted as a critical factor by the participants. The importance and influence that governance has on effective integration in mental health care has been previously identified [57]. Indeed, pharmacists' integration in primary care was also identified as a requirement for the provision of mental health care in community pharmacies. However, pharmacists and pharmacy staff in metropolitan areas reported not having a close relationship with other healthcare professionals and not being fully considered part of the primary care team. Recent research regarding community pharmacists' integration within primary care in Australia stated the need to include policy and funding support to promote integration models and enable access to services conducted in conjunction with pharmacists [58].

Good governance has also been associated to improvements in the safety and quality of health care services through the implementation of policy, educational materials and processes for improvement. It determines how health services are delivered and has a direct influence on the safety and quality of services [59]. Consideration should be given to the policy and legislative changes required to further regulate, define and promote the role of community pharmacists in mental health care and as an integral part of primary care teams. The framework proposed as a result of this study may be a first step to strengthening governance around mental health support in community pharmacies. This in return, maybe essential to promote consistency and quality assurance of the services provided by community pharmacists to people dealing with mental health conditions. The use of guides or structured protocols to standardise mental health services delivery and protocolise how community pharmacists and staff approach people living with mental illness, as proposed by participants in this study, appears to be a feasible solution to increase the quality of service provided while ensuring consumers receive consistency in service and referral.

Another factor identified as a barrier by participants was the lack of support in the workplace influenced by others such as lack of remuneration and time. A similar situation was found in a study exploring community pharmacists' perception of their role in primary mental health care where participants reported the support provided to consumers was influenced by the philosophy of the business owner [60]. This is not surprising as service provision has been previously reported as time-demanding, making it challenging to deal with other commitments in the pharmacy [29]. Furthermore, this situation is even less sustainable by the lack of remuneration, preventing community pharmacists from providing more comprehensive services [61]. Thus, to guarantee that community pharmacists have the capacity to provide mental health support and that this support is sustainable, the development of a specific funding model should be considered.

Privacy was identified as a barrier for community pharmacists in rural and regional areas. Consumers located in smaller population density areas appeared to have a bigger reticence to disclose personal information. This may be attributable to the fear that this information will pass to their community. Having a private area to provide professional services in the community pharmacy has been proven to help consumers build trusting relationships with their pharmacists [62]. Thus, having a consultation room within the community pharmacy seems to be essential to enhance consumers' willingness to receive mental health services.

Most participants in urban and regional areas reported that the primary contact of referral was GPs and, in the most severe cases, to mental health crisis teams. Published research focused on pharmacist and GP collaboration has shown benefits (i.e., improved drug knowledge, sharing of care and clinical reassurance when managing complex consumers, easy access to consumer information, better integration, and satisfaction) [43, 63, 64]. Participants also indicated they referred consumers to online and phone mental health services for consumers with milder mental illness. These services appear to positively benefit consumers who are unwilling to use other available services and those with limited access [65]. Providing information around these services and options at the community pharmacy appears to be an opportunity, for those not already doing so, to ensure consumers have access to tools to deal with their mental illness.

Participants suggested ideas to improve the delivery of services by community pharmacists in mental health. Regardless of having undertaken the Mental Health First Aid course, participants commented on the possibility of including additional training with more practical scenarios relevant to community pharmacy and using people with lived experience. A Blended version of the MHFA course specifically tailored for pharmacists and pharmacy staff is currently available. This version consists of self-paced online learning modules followed by a practical classroom-based (face-to-face or live webinar) session using case-studies, videos and resources tailored to their learning needs [66]. However, the standard face-to-face MHFA course seems to be still the most common version used by pharmacists [67]. Pharmacists' [68] and pharmacy students' skills after receiving MHFA training have been assessed in the literature (e.g., by using simulated consumer scenarios) [50] and have demonstrated the positive influence of training on pharmacists' knowledge, attitudes, and confidence [50, 51, 68].

The use of real-life scenarios and the inclusion of multidisciplinary teams during training have shown improvements in the skills of healthcare professionals [69]. In the same way, studies including role-play as part of the training have demonstrated improved professional skills (e.g., recognition of possible scenarios and solutions, increased confidence and communication skills, promoted effective discussion, and active participation) [70, 71]. Despite this, some studies assessing pharmacists' confidence to provide mental health services using self-assessment tools have identified that participants often overestimate their confidence to deliver the services in practice after training [72, 73]. Nonetheless, including people with life experiences as part of the training to simulate real life situations should be considered to enhance pharmacists' mental health skills in practice.

Participants highlighted the need to promote their role in mental health and agreed that it would be beneficial to promote services offered by community pharmacies to increase community awareness. One study conducted by da Costa et al. demonstrated that a pharmacists' awareness campaign about early detection of atrial fibrillation enhanced effective communication pathways for interprofessional collaboration [74]. Greater familiarity and comfort with available mental-health resources may help alleviate some of the barriers that community pharmacists experience concerning mental health pharmacy practice. Participants also identified the need for remuneration for service provision and suggested the possibility of funding a medication review service (e.g., MedsCheck) focused on mental health. The effectiveness of

medication review services in supporting consumers with chronic diseases have been widely reported in the literature [75, 76]. Specifically, a study conducted by McMillan et al. showed that mental health medication support service delivered by trained pharmacy staff in community pharmacies across Australia had a positive impact on consumers outcomes [77]. This should be considered to enhance the care of consumers with mental health conditions whilst ensuring the continuity and sustainability of these services over time.

As part of the present study, a framework to guide pharmacists' and consumers' empowerment in mental health is proposed. This model depicts two main domains (pharmacists, consumers). Both domains are influenced by external factors and individual factors (i.e., barriers and facilitators moderating pharmacists'/ consumers' empowerment). The most influential factor to enable pharmacists' roles in mental health and integration within primary care is governance. Having a structured system supporting the delivery of mental health services in community pharmacies is required as a first step to enhance and define the role of pharmacists' as mental health providers. Additionally, funding is required to facilitate the provision of mental health services and guarantee their sustainability. Lack of communication and integration with other primary care providers promotes pharmacists' confusion about their role and where they fit within the primary care team, which can also influence pharmacists' confidence in providing these services. A workplace that supports and encourages pharmacists and pharmacy staff to grow professionally is essential to successfully delivering this type of service. Moreover, dealing with people with mental issues can negatively impact pharmacists' own mental health, which affects their receptiveness and willingness to provide mental health services to the community. Pharmacists' self-empowerment is determined by their skills and knowledge. Continuous training and education are required to adapt to new situations, respond to different scenarios and provide successful services.

Consumers' empowerment is influenced by personal factors such as their knowledge and health literacy. Consumers' understanding of their condition is a facilitator in their awareness and motivation to independently seek help. Pharmacists' and pharmacy staff skills such as empathy, reassurance, verbal and non-verbal communication are crucial to reduce consumer stigma and create a supportive relationship with consumers. This may also influence the consumers' willingness to change and increase their confidence to discuss their conditions and medications. Lastly, pharmacist and consumer empowerment in mental health are also affected by characteristics of the community pharmacy. The physical layout of the pharmacy and having a private area to provide these services may influence consumers' comfort and confidence to discuss their condition.

## Strengths and limitations

Thirty-three community pharmacists and pharmacy staff from urban, regional and rural areas of NSW, Australia participated in focus groups. The findings of this research may not represent the full spectrum of opinions/experiences of community pharmacists across Australia. Therefore, their replicability to community pharmacies located in other areas may be indicative but limited. The use of NGT facilitated the needed structural framework to discuss and obtain via consensus a list of the most significant barriers and facilitators moderating the delivery of mental health services in community pharmacy. However, the NGT approach can be regimented and lends to a single purpose. Thus, open discussion was conducted to allow more in-depth exploration of participants' ideas. A limitation of this study is that the framework proposed includes factors moderating consumers' empowerment in mental health from pharmacists and staff perspectives. Therefore, this should be considered as some of the factors identified may differ from future research including customers' perspectives.

## Conclusion

The qualitative work undertaken in this study explored community pharmacists' and pharmacy staff experiences and perspectives regarding the provision of mental health services in community pharmacy. Overall, community pharmacists and pharmacy staff role in mental health is moderated by several factors. The exploration of referral pathways used by community pharmacists to refer a person living with a mental illness has made evident the lack of integration of community pharmacists within mental health primary care pathways, recognition of the pharmacist in the management of mental health, and limitation of service delivery through limited remuneration. These findings are consistent with the wider body of research within the discipline, in other clinical areas. As a result of this research, a framework detailing the factors for community pharmacists and pharmacy staff influencing the delivery of mental health services in pharmacy is proposed. Future research should assess the applicability of the framework in practice in other settings. Furthermore, customers perspectives and experiences regarding mental health service delivery in community pharmacies should be explored to validate the results of this study and identify additional factors affecting their empowerment.

## Supporting information

**S1 Appendix. Factors moderating pharmacists and consumers' empowerment in mental health and supporting quotations resulting from the thematic analysis in Round One.**
(DOCX)

**S2 Appendix. Factors highlighted for pharmacists' empowerment in mental health and supporting quotations resulting from the thematic analysis in Round-Two.**
(DOCX)

## Acknowledgments

We would like to thank and acknowledge the pharmacists and pharmacy staff who agreed to participate. Without their commitment to participating, this project would not have been possible.

## Author Contributions

**Conceptualization:** Sarah Dineen-Griffin, John Rae, Rodney A. Hill.

**Data curation:** Carmen Crespo-Gonzalez, Sarah Dineen-Griffin, John Rae.

**Formal analysis:** Carmen Crespo-Gonzalez, John Rae.

**Funding acquisition:** Rodney A. Hill.

**Investigation:** Carmen Crespo-Gonzalez, Sarah Dineen-Griffin, John Rae.

**Methodology:** Carmen Crespo-Gonzalez, Sarah Dineen-Griffin, John Rae.

**Project administration:** Rodney A. Hill.

**Resources:** Rodney A. Hill.

**Supervision:** Rodney A. Hill.

**Writing – original draft:** Carmen Crespo-Gonzalez.

**Writing – review & editing:** Carmen Crespo-Gonzalez, Sarah Dineen-Griffin, John Rae, Rodney A. Hill.

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
