## [Decision Letter · Decision Letter 0]

22 Dec 2021

PONE-D-21-35278A qualitative exploration of mental health services provided in community pharmaciesPLOS ONE

Dear Dr. Hill,

Thank you for submitting your manuscript to PLOS ONE. After careful consideration, we feel that it has merit but does not fully meet PLOS ONE’s publication criteria as it currently stands. Therefore, we invite you to submit a revised version of the manuscript that addresses the points raised during the review process.

We look forward to receiving your revised manuscript.

Kind regards,

Vijayaprakash Suppiah, PhD

Academic Editor

PLOS ONE

Journal Requirements:

Reviewers' comments:

Reviewer's Responses to Questions

**Comments to the Author**

1. Is the manuscript technically sound, and do the data support the conclusions?

Reviewer #1: Yes

Reviewer #2: Yes

Reviewer #3: Yes

2. Has the statistical analysis been performed appropriately and rigorously? 

Reviewer #1: Yes

Reviewer #2: N/A

Reviewer #3: N/A

3. Have the authors made all data underlying the findings in their manuscript fully available?

Reviewer #1: Yes

Reviewer #2: Yes

Reviewer #3: Yes

4. Is the manuscript presented in an intelligible fashion and written in standard English?

Reviewer #1: Yes

Reviewer #2: Yes

Reviewer #3: Yes

5. Review Comments to the Author

Reviewer #1: Title: A qualitative exploration of mental health services provided in community pharmacies

Comments to the Author

Thank you for the opportunity to review this manuscript. In this paper, the authors present the results of a qualitative study using thematic analysis based on data collected from community pharmacists and pharmacy staff across metropolitan, regional, and rural areas of New South Wales, Australia. The objectives of this study are to examine the factors that support the delivery of mental health services in Australian community pharmacies and propose ideas for improvement, particularly in regional and rural regions.

Major comments

Introduction:

The background provides context for this research; however, I think that it is a bit long and unfocused. Specifically, it covers mental health problems globally and in Australia and their negative impacts on the economic side, and how COVID-19 deteriorates mental health problems. In my opinion, the real meat of the introduction begins on Page.5 line 104 because this is the introduction of the roles of community pharmacies in delivering mental health care in Australia. I think that the authors should re-assess the material prior to this paragraph and determine which information is most germane to meet this study's primary objective. In addition, I think the authors should highlight the novel of this study.

Methods:

Major Comment 1:

There is a lot of duplication of methodology between the text and figure 1. I would encourage the authorship team to streamline presentation of information. I would also encourage authors put data analysis section together if applicable.

Major Comment 2:

The authors state participants were recruited using purposive sampling of focus groups. Please elaborate in greater detail on how this was done, including 1) What are the inclusion and exclusion criteria for recruitments? 2) Do authors include chain pharmacies or independent pharmacies? 3)Do those pharmacies offer any specialty services (e.g., mental health services) before recruiting into the study? 4) Do those pharmacies have similar volume of prescription dispensed per day/week or have similar pharmacist/staff ratios? Elaboration on the above details will give readers greater insight an assurance that the integrity of data (and subsequent conclusion drawn from them) are defensible

Major Comment 3:

Line 158. The author state that “Each focus group meeting was divided into two stages” in Round one. However, only the first 4 focus group went through 2 stages based on the context and figure 1. Please clarify if there is a discrepancy existing between the statement and figure 1 or provide more information or purposes why choose to use different methodology in Round 1? In addition, consider adding the aim of the Round 1

Major Comment 4:

Line 242. The author state that “The goal was to recruit between 4 and 7 participants per focus group” in Round 2. The number of participants in Round 2 per group was less than Round 1 and the number of focus group was also less than Round 1. Would it be a potential bias in data reporting or validation?

Results:

Major Comment 1:

Author mentioned that there is a network of 5,822 pharmacies in Australia in the introduction section. Would the results of this study have any potential bias because of the sample size?

Major Comment 2:

The authors identified 24 factors that enable or hinder mental health service delivery in community pharmacies. It is a little bit unclear how did those factors be identified? Were they identified through the 2 nominal questions in Round 1 only or combined with thematic analysis later? Consider elaborating in greater detail on how this was done.

Major Comment 3:

What were the themes from Round 1 and Round 2? I would encourage the authors to use more tables to summarize the themes with supportive quotes from Round 1 and Round 2. It is good to have great details in the texts, but sometimes it is hard to follow, especially since this manuscript has two rounds of information with multiple themes. Therefore, I would recommend streamlining some texts and putting the additional quotes into tables if applicable.

Discussions:

Major Comment 1:

Paragraph 5. The authors discussed the impact of governance in this paragraph. Per table 5, governance is defined as the availability of resources to guide, support, maintain and improve the reliability, safety, and quality of the services and standardise mental health service delivery in community pharmacy. There is no clear relationship between governance and policy. Consider providing more details about the governance in the text and how it links to the policy in the discussion.

Major Comment 2:

Limitation. The authors should state generalizability as a limitation due to the sample size. Generalizability may further limit the methodology which considers the experiences of participants which may vary considerably among different pharmacies. In addition, selection bias may be existing because some pharmacists were certified and some were not.

Conclusion:

Major Comment 1:

In the method section, the authors did describe the plan to evaluate the role of pharmacies during COVID-19 pandemic. To me, it is a bit inappropriate to draw the conclusion because it was not derived from this work in the results section.

Additional note:

1. Figures: I would recommend increasing the resolutions of all figures

2. Line 255. The author state that “participants were presented with a framework”. It is a little bit unclear how the framework was presented. Did participants see the figure 2?

3. Table 1. The table shows that some pharmacists and pharmacy staff who were not Mental Health First Aid certified. Is this a nationwide or statewide level certification? Would the results be different if study only included pharmacists and pharmacy staffs who are Mental Health First Aid certified?

4. Line 592. Missing figure# or word in the beginning?

5. Consider to include what is the next step of the future research to expand the role of community pharmacy in mental health delivery in Australia

Reviewer #2: This is a small study, and reflects a potential trend?...I would widen the population of community pharmacists to at least statistical significance, and then present the date as a potential alternative therapeutic modality which may supplant others.........

The big error that I find is thst this only addresses prescription meds, and not OTC meds which are 'prescribed' by the pharmacists.......that of course is an established mode, but some are also of precriptive use if dose is doubled.....many are OTC at 1/2 dose.....hmmmmm...

Reviewer #3: This paper describes a round of focus groups with pharmacists and pharmacy staff exploring facilitators and barriers to mental health services in community pharmacy. It addresses an important area of trying to progress much needed support for pharmacists’ roles in mental health care. It is generally well written but I have two key issues with the paper as present:

• Length – although I note that the submission guidelines do not stipulate a word count, I feel this paper is excessively long and the paper could be significantly shortened, almost be half. The key messages get lost and a succinct paper is much clearer for the reader. I estimated the word count to be more than 8700 words.

• The proposed framework suggests factors moderating patients’ (note comment below about language and use of term patient) self-empowerment. However, I have major concerns about this part of the framework as consumers/people with lived experience were not part of this research. Data was only gathered from pharmacists/pharmacy staff and I feel it is too much of a reach to then propose what moderates consumers self-empowerment based on the data. I suggest framework be revised to acknowledge this and identify consumer involvement in this area as where further research is needed.

Background:

• It is currently very long and could be significantly cut down. For example, the first 1.5 pages could be condensed into 1 paragraph about the prevalence and burden of mental health problems

• A variety of references are used for these figures on prevalence/burden. Ensure the primary source is being cited – e.g. Black Dog Institute flyer rather than data from the National survey of mental health and wellbeing

• Introduction to pharmacists’ roles in mental health describes what pharmacists can do, but it should summarise the evidence supporting these roles. In particular, what is known in the literature about barriers and facilitators to pharmacists’ roles in mental health care already? What governance/policy frameworks already exist for the pharmacy profession in the mental health space? In addition, what is known about barriers and facilitators to pharmacists’ roles more broadly in other areas that could be adapted here? The introduction is very long, but it could be much more succinct as well as better at identifying the gaps in the literature and how this study aims to fill those gaps.

Methods:

• I feel the methods a little confusing to follow. Round one has stage one and stage two with 4 focus groups using NGT, then a further two focus groups. These seems like two separate phases, and it seems like three rounds of focus groups not two?

• Page 9 line 224 – This sentence is a bit confusing and could be clarified? ‘to identify the underpinning reasons for participants selection and examine the data obtained’

• Page 10 line 244 – interview guide or focus group guide?

Results:

• ‘allocated according to their characteristics into six groups’ – explain how allocation occurred? And clarify did this mean that participants with training or no training were grouped together into focus groups?

• Table 1 – acronyms used here and in test of results. These are confusing to follow as reader needs to keep going back to see what they mean. I question whether they are useful?

• Table 2 – some of the figures don’t add up. E.g. pharmacy staff – 6 born in Aust and 2 born in England but only 7 in total? For country of birth – if only 1 or 2 in non-Aust born category perhaps merge?

• How do demographics compare to pharmacist population across Australia? Seems to be very high number of females and percentage born in Australia which may not be generalizable with pharmacy population.

• Page 17 – line 307 onwards. Thematic analysis referred to here – can it be clarified where this data comes from? I think it is stage two of round one? But it is confusing to follow

• Page 17 – line 319 – ‘mental health specialist’ – language here is incorrect/inappropriate. Pharmacists cannot directly refer to a specialist such as a psychiatrist. Rather it seems the pharmacists are referring to calling an ambulance in a mental health crisis or referring back to a community mental health team, which has nurses/psychiatrists as part of their mental health care team.

• Should be consistent with describing demographics of participants in round one and two. Round two, readers only told that all are female? Perhaps merge these sections to the start of results to clarify number and demographics of all participants in each round – then go on to describe results?

• Page 22 – first sentence. Clarify grammar

• General comment re quotes in results section. I suggest cutting down the number and length of quotes. A number of quotes are duplicative and others are very long which could be cut down to just present key points. E.g. quotes of page 24 could be cut down to one on guidelines and one on governance but currently there are 8 quotes. There are numerous examples of this that could help cut down the length of this paper.

• I don’t think it is appropriate to name pharmacy chains – deidentify these. Also, these are not relevant to an international audience.

• Page 26 – line 528 – grammar

• Add footnotes to clarify any Australian relevant things in quote which need context for an international readership – e.g. medicare item number on page 28

• Table 5 – using wheelchair example of accessibility doesn’t seem to fit with what is often meant by accessibility here? Ie being able to talk to a pharmacist 7 days a week, without an appointment etc?

Discussion:

• Both the introduction and discussion mention the COVID-19 pandemic, but this does not seem to be mentioned in any of the results? Particularly as this is how the discussion starts? While the literature describes the additional pressures placed on communities as well as pharmacists in the mental health space, the results here don’t describe this. I would suggest removing discussion relating to the pandemic, except perhaps to highlight the even greater role pharmacies have had to play because of the pandemic.

• The discussion is generally well written but there is significantly more literature known in the area of the effectiveness of pharmacist led mental health services – even systematic reviews that should be cited/discussed to give context of what is known in the area and how these results add value. As well as literature on barriers/facilitators to mental health service delivery.

• In addition, literature outside the pharmacy space should be discussed/compared to look at what is known that could contribute here. E.g. in the area of mental health stigma, patient empowerment. Do consumers even want pharmacists to have a role in mental health service delivery?

• Page 37 – MHFA – give reference and definition of what this is as not necessarily known to all. Or when mentioning a service such as MedsCheck – add footnote or a reference.

• Discussion around MHFA. There is a pharmacy specific version of MHFA available which has pharmacy content and case studies. There is also a refresher course available already where participants can get re-accredited after a 3-year period. There is also significant literature in the pharmacy space about MHFA evaluation, the use of simulation to test MHFA skills and confidence. These should be discussed here.

• Limitations – I suggest adding a limitation about the lack of consumer involvement in this study.

General:

• Language in mental illness. A few terms used in the paper can be problematic or stigmatizing when describing people living with a mental illness. People often do not like to be considered a ‘patient’ and you may consider an alternative term, e.g. a person with lived experienced of/or living with a mental illness, or mental health consumer. Also, the term ‘suffer’ has negative connotations and it is preferred to describe as ‘living with’ or ‘experiencing’

6. PLOS authors have the option to publish the peer review history of their article (what does this mean?). If published, this will include your full peer review and any attached files.

Reviewer #1: No

Reviewer #2: **Yes: **Gerald Dieter Griffin,PharmD,MD

Reviewer #3: No

---

## [Author Response · Author response to Decision Letter 0]

5 Feb 2022

All responses have been included within the Response to Reviewers document.

---

## [Decision Letter · Decision Letter 1]

28 Mar 2022

PONE-D-21-35278R1A qualitative exploration of mental health services provided in community pharmaciesPLOS ONE

Dear Dr. Hill,

Thank you for submitting your manuscript to PLOS ONE. After careful consideration, we feel that it has merit but does not fully meet PLOS ONE’s publication criteria as it currently stands. Therefore, we invite you to submit a revised version of the manuscript that addresses the points raised during the review process.

We look forward to receiving your revised manuscript.

Kind regards,

Vijayaprakash Suppiah, PhD

Academic Editor

PLOS ONE

Journal Requirements:

Reviewers' comments:

Reviewer's Responses to Questions

**Comments to the Author**

1. If the authors have adequately addressed your comments raised in a previous round of review and you feel that this manuscript is now acceptable for publication, you may indicate that here to bypass the “Comments to the Author” section, enter your conflict of interest statement in the “Confidential to Editor” section, and submit your "Accept" recommendation.

Reviewer #1: All comments have been addressed

Reviewer #3: (No Response)

2. Is the manuscript technically sound, and do the data support the conclusions?

Reviewer #1: Yes

Reviewer #3: Yes

3. Has the statistical analysis been performed appropriately and rigorously? 

Reviewer #1: Yes

Reviewer #3: N/A

4. Have the authors made all data underlying the findings in their manuscript fully available?

Reviewer #1: Yes

Reviewer #3: Yes

5. Is the manuscript presented in an intelligible fashion and written in standard English?

Reviewer #1: Yes

Reviewer #3: Yes

6. Review Comments to the Author

Reviewer #1: Title: A qualitative exploration of mental health services provided in community pharmacies

Comments to the Author

Thank you for the opportunity to review this revised manuscript. In this version, the authors made major revisions to response reviewers’ comments. I only have a few comments listed below and I hope they are useful to the authorship team.

My comments are located

Major comments

Introduction:

Consider removing the first paragraph and start with the current second paragraph to help readers knowing this is a study conducted in Australia sooner.

Consider combining the third and fourth paragraph together (or remove the fourth paragraph) because the fourth paragraph does not provide new essential information to readers after reading the previous paragraph.

Line 82. A typo. C should be lower case.

Line 103. Remove “community”

Line 109. Consider adding New South Wales (NSW) here so that readers would know this is not a nationwide qualitative study.

Methods:

Line 120. Consider adding New South Wales (NSW) here so that readers would know this is not a nationwide qualitative study.

Table 3. Not clear what does “Facilitator identified in urban and regional/rural areas” mean here. Do authors go back, and check where did each participant from?

Table 4 Not clear what does “Barrier identified in urban and regional/rural areas” mean here. Like the comment for Table 3.

Results:

No further comments for results section.

Discussions:

No further comments for discussions section.

Conclusion:

No further comments for conclusion section.

Additional note:

1. Figures: I would recommend increasing the resolutions of all figures. They are still not very clear.

Reviewer #3: Very minor comments. The authors have a done a good job at addressing reviewer comments. Only a few minor points left:

Grammar –

• still see patient used instead of consumer or person a few times (e.g. in abstract). Authors to check and ensure consistency.

• on page 61 line 189. Is ‘December of 2020’ – just ‘December 2020’

• Page 61 line 193 – after text removed – sentence is very brief?

• Ensure acronyms spelled out first in text – ie MHFA. It is spelled in table 1 but not in text

FG questions now in figure. This is better but perhaps text can summarise what they are about – e.g. on page 63 line 239-242 as this paragraph doesn’t say point of

There are now a lot of tables. Participant demographics from round one and two could be merged? Ie with additional columns

I couldn't see figures attached in revision. I assume these remain unchanged from version 1

7. PLOS authors have the option to publish the peer review history of their article (what does this mean?). If published, this will include your full peer review and any attached files.

Reviewer #1: No

Reviewer #3: No

---

## [Author Response · Author response to Decision Letter 1]

1 Apr 2022

All editor requests have been addressed in the accompanying materials

---

## [Decision Letter · Decision Letter 2]

26 Apr 2022

A qualitative exploration of mental health services provided in community pharmacies

PONE-D-21-35278R2

Dear Dr. Hill,

We’re pleased to inform you that your manuscript has been judged scientifically suitable for publication and will be formally accepted for publication once it meets all outstanding technical requirements.

Kind regards,

Vijayaprakash Suppiah, PhD

Academic Editor

PLOS ONE

Reviewers' comments:

Reviewer's Responses to Questions

**Comments to the Author**

1. If the authors have adequately addressed your comments raised in a previous round of review and you feel that this manuscript is now acceptable for publication, you may indicate that here to bypass the “Comments to the Author” section, enter your conflict of interest statement in the “Confidential to Editor” section, and submit your "Accept" recommendation.

Reviewer #1: All comments have been addressed

2. Is the manuscript technically sound, and do the data support the conclusions?

Reviewer #1: Yes

3. Has the statistical analysis been performed appropriately and rigorously? 

Reviewer #1: Yes

4. Have the authors made all data underlying the findings in their manuscript fully available?

Reviewer #1: Yes

5. Is the manuscript presented in an intelligible fashion and written in standard English?

Reviewer #1: Yes

6. Review Comments to the Author

Reviewer #1: (No Response)

7. PLOS authors have the option to publish the peer review history of their article (what does this mean?). If published, this will include your full peer review and any attached files.

Reviewer #1: No

---

## [Editor Report · Acceptance letter]

3 May 2022

PONE-D-21-35278R2 

A qualitative exploration of mental health services provided in community pharmacies 

Dear Dr. Hill:

I'm pleased to inform you that your manuscript has been deemed suitable for publication in PLOS ONE. Congratulations! Your manuscript is now with our production department. 

Kind regards, 

on behalf of

Dr. Vijayaprakash Suppiah 

Academic Editor

PLOS ONE